# Bisphenol A Exposure and Sperm ACHE Hydroxymethylation in Men

**DOI:** 10.3390/ijerph16010152

**Published:** 2019-01-08

**Authors:** Xiuxia Song, Maohua Miao, Xiaoyu Zhou, Dekun Li, Youping Tian, Hong Liang, Runsheng Li, Wei Yuan

**Affiliations:** 1Key Laboratory of Reproduction Regulation of NPFPC, SIPPR, IRD, Fudan University, Shanghai 200237, China; xxsong1212@163.com (X.S.); miaomaohua@sippr.org.cn (M.M.); 16211150003@fudan.edu.cn (Y.T.); lianghong@sippr.org.cn (H.L.); 2Shanghai Tongshu Biotechnology Co., Ltd., Shanghai 200237, China; zouxy2001@163.com; 3Division of Research, Kaiser Foundation Research Institute, Kaiser Permanente Northern California, Oakland CA 94612, USA; dkl@dor.kaiser.org; 4School of Public Health, Fudan University, Shanghai 200237, China

**Keywords:** bisphenol A, ACHE, DNA hydroxymethylation, sperm

## Abstract

Exposure to bisphenol A (BPA) has been shown to impact human sperm quality. The epigenetic mechanisms underlying the effect remain unknown. The acetylcholinesterase (ACHE) gene is a sperm-expressed gene encoding the acetylcholine hydrolyzing enzyme acetylcholinesterase and participates in the apoptosis of cells, including sperm. This study aimed to examine whether BPA exposure is associated with the hydroxymethylation level of the sperm *ACHE* gene. A total of 157 male factory workers were studied, among whom 74 had BPA exposure in the workplace (BPA exposure group) and 83 had no BPA exposure in the workplace (control group). Urine samples were collected for BPA measurement and semen samples were collected to assay for *ACHE* hydroxymethylation. Sperm *ACHE* hydroxymethylation level was higher in the BPA exposure group (*p* = 0.041) compared to the control group. When subjects were categorized according to tertiles of detected BPA level, higher *ACHE* hydroxymethylation levels were observed for the lowest, middle, and top tertiles compared to those with BPA below the limit of detection (LOD). In a linear regression analysis adjusted for confounders, a positive linear association between urine BPA concentration and 5-hydroxymethylcytosine (5hmC) rate of the sperm *ACHE* gene was observed, although the association did not reach statistical significance in all categories after being stratified by the BPA tertile. In conclusion, 5hmC of the sperm *ACHE* gene was positively associated with BPA exposure, which may provide supportive evidence for BPA’s effects on male fertility or other health endpoints.

## 1. Introduction

Bisphenol A (BPA) is a synthetic industrial chemical that has been widely used in the manufacturing of polycarbonate plastic and epoxy resins, such as water and baby bottles, reusable food and drink containers and some dental sealants. BPA can enter the body through diet, the dermis, and the respiratory tract [1,2,3]. Exposure to BPA has gained wide attention over the past decades due to its ubiquitous exposure and potential endocrine disrupting effects, including weak estrogenic and strong anti-androgenic and anti-thyroid activities [4,5,6,7]. An increasing body of evidence has shown that exposure to BPA is associated with higher risks of adverse health effects in humans, including cardiovascular disease, insulin resistant diabetes, obesity, and cancers [8,9,10,11]. Particularly, BPA exposure is associated with impaired male reproductive functions, including reduced semen quality, altered reproductive hormones, and decreased sexual function [12,13,14,15,16]. However, the mechanisms underlying these effects remain to be elucidated.

Epigenetic mechanisms have been demonstrated to play an important role in BPA’s biological effects, including the effects on spermatogenesis [17,18]. For instance, long interspersed element-1 (LINE-1), the most abundant and the only active autonomous non-long terminal repeats (non-LTR) retrotransposon in the human genome, showed an alteration in methylation following BPA exposure [19]. However, little is known about how BPA modulates DNA demethylation. An essential intermediate of active DNA demethylation processes, 5-hydroxymethylcytosine (5hmC), which is oxidized from 5-methylcytosine (5mC) by the ten-eleven translocation (TET) family of proteins, has been reported to modulate the demethylation associated with BPA [17,20,21]. A recent animal study reported that exposure to BPA inhibits global DNA hydroxymethylation in the adult testis and decreases testicular TETs [22]. More importantly, in our recent study based on pooled human sperm, the total level of 5hmC increased significantly in subjects with BPA exposure [23]. 5hmC may be another unique and dynamic marker of DNA demethylation regulation [24,25].

The acetylcholine hydrolyzing enzyme acetylcholinesterase (ACHE) is conventionally known for terminating cholinergic neurotransmission. However, non-cholinergic roles of ACHE are observed in animal and human sperm [26,27]. ACHE has been shown to be a potential marker and regulator of apoptosis in some cells [28,29,30]. Studies have also revealed the functions of ACHE during spermatogenesis, including interactions with the receptor of activated protein kinase C (RACK1) to promote apoptosis and with the glycolytic enzyme enolase-α, increasing enolase activity, to reduce sperm differentiation and sperm counts [31,32]. Our recent study based on pooled samples found that the 5hmC levels of promoter regions in sperm-expressed genes (including *ACHE* gene) are significantly higher than in sperm-repressed genes and that the *ACHE* gene is a sperm-expressed gene [23]. The objective of our study was to examine the association between BPA and 5hmC in the sperm *ACHE* gene in individuals, which may help us understand the mechanism of BPA’s effects on male fertility.

## 2. Materials and Methods

### 2.1. Study Population

Male participants came from a prospective cohort study that has been described in detail elsewhere [12,19,33]. In brief, 74 males were recruited from factories that manufacture BPA and epoxy resin in three regions (Ningbo, Wuxi, and Yueyang) of China from 2004 to 2008 (BPA exposure group) and 83 males from factories without occupational BPA exposure in the same region during the same period (control group). The study was approved by the ethics committee board of the Shanghai Institute of Planned Parenthood Research (IRB00008297), and all participants signed informed consent before participating in the study.

### 2.2. Data and Biosamples Collection

Information on socio-demographic characteristics (age and education), lifestyle factors (smoking and alcohol consumption), and history of disease (any acute or chronic disease of the liver, kidney, or other organs) were collected through in-person interviews by trained interviewers. Semen and urine samples were also collected at the same time using the methods previously described [12,19].

### 2.3. BPA Measurement

BPA can be measured in various biological samples. Although plasma, urinary, and seminal BPA are correlated with each other [34,35,36], urinary BPA is most widely used in epidemiological studies [37]. In the present study, we only measured urinary BPA concentration of the workers. We did not have measurements of BPA levels in semen. Two urine samples (pre-shift and post-shift) were collected from each male worker of the BPA-exposed group, and one urine sample was collected from those of the control group. For each urine specimen, modified high-performance liquid chromatography (HPLC) was utilized to measure the total urine BPA concentration (free plus conjugated species), as previously described [38]. Briefly, urine samples were treated with phosphorous acid buffer/β-glucuronidase for hydrolyzation and were subsequently extracted twice using ether (HPLC grade, Dikma, Foothill Ranch, CA, USA). The supernatants were collected and evaporated with nitrogen gas. The residue was dissolved in 60% acetonitrile and analyzed using HPLC equipment. The limit of detection (LOD) was 0.31 μg/L. Creatinine-corrected (μg/g creatinine) BPA concentration was used in the analyses to account for urine dilution. To better represent the actual BPA exposure levels for the BPA-exposed group, the mean BPA concentrations of the pre-shift and post-shift samples were used in the analysis.

### 2.4. DNA Extraction and DNA Hydroxymethylation Analysis

The sperm DNA was prepared as described previously [19]. Briefly, sperm specimens were treated by guanidine hydrochloride and sodium citrate to isolate sperm pellets, and then precipitated with ethanol. Then, sperm DNA was isolated by a standard phenol/chloroform purification method and qualified by electrophoresis on an agarose gel and visualized with ethidium bromide.

5hmC was analyzed and quantitated using the EpiMark 5hmC and 5mC Analysis Kit (NEB, #E3317S) according to the manufacturer’s protocols. Briefly, DNA was first mixed with UDP-glucose, then split into two parts that were incubated with or without T4 beta-glucosyltransferase (T4-βGT), respectively, for 16 h at 37 °C. This glucosylation was followed by restriction endonuclease digestion. Both reaction mixtures were run in triplicate and were mock digested for at least 4 h with MspI or with HpaII. Samples were treated with proteinase K and incubated at 40 °C for 30 min. Proteinase K was then inactivated by incubating at 95 °C for 10 min. The fraction of glycosylated DNA and, therefore, protected MspI sites, as well as the fraction of 5mCand 5hmCsensitive sites (determined using HpaII restriction) at *ACHE* gene loci, were quantified by quantitative polymerase chain reaction (qPCR) using primers (F-ATGCAGTGACAGGCACAGAC, R-TGAGTGTCCCACGTCACCTTT). The rate of hydroxymethylation was calculated using the formulae in the kit according to the manufacturer’s protocols.

### 2.5. Statistical Analysis

The distributions of creatinine-corrected BPA concentrations by subjects’ characteristics were tabulated. The urine BPA concentrations were natural log (ln) transformed to achieve a normal distribution and then an independent *t*-test or one-way ANOVA analysis was performed to analyze differences in BPA levels. When the concentration of BPA was below the limit of detection, a value of LOD/√2 was employed. To make the results more interpretable, we classified subjects with detected urine BPA levels into three categories. Therefore, all subjects were divided into four groups: BPA undetected (lower than the LOD level), lowest tertile, middle tertile, and top tertile. Means (SD) and 5th, 25th, 50th, 75th, and 95th percentiles of 5hmC rates of sperm *ACHE* gene were used to describe the distribution of 5hmC by occupational BPA exposure (yes or no) and the urine BPA levels. Rates of 5hmC were then natural log (ln) transformed to achieve a normal distribution for the linear regression analysis, which was used to examine the association between categorized BPA exposure and 5hmC. Age, smoking, alcohol consumption, and history of disease, which had been reported to be associated with 5hmC [39,40,41], were adjusted for as potential confounders.

We also conducted a linear regression analysis to examine the linear association between BPA exposure and 5hmC using continuous BPA level (log transformed). We repeated the analyses within each tertile of urine BPA to examine whether the association varied by exposure dosage. Subgroup analyses were also conducted in the non-occupational exposed group, results of which reflect the effect of environmental low dose BPA exposure and would corroborate the findings to some extent. All statistical analyses were performed using SPSS 19.0 software package (IBM SPSS, Armonk, NY, USA).

## 3. Results

The detection rates of urinary BPA levels in the exposure group and control group were 100% and 36.14%, respectively. The mean BPA concentrations in the pre-shift and post-shift samples were used in the exposure group. The difference and distribution of BPA levels between the pre-shift and post-shift exposure groups are presented in a supplemental table (Appendix A). BPA levels by the characteristics of the study subjects are presented in Table 1. Compared to men with middle school or below, BPA levels in those with high school and with college and above were relatively lower (Appendix A). The most important determinants for a high BPA level in this study population was occupational exposure. The geometric mean (GM) of BPA concentration in the occupational BPA exposure group was significantly higher than that of the control group (199.13 μg/g Cr vs. 0.77 μg/g Cr). The GM of BPA levels were 3.93 μg/g Cr, 42.58 μg/g Cr, and 2937.79 μg/g Cr for the lowest, middle, and top tertiles, respectively. Urine BPA levels were not associated with age, history of chronic diseases, smoking, and alcohol consumption.

The distribution of 5hmCof the sperm *ACHE* gene by occupational BPA exposure (yes or no) and categories of urine BPA levels are described in Table 2. The 5hmC rate of the sperm *ACHE* gene was higher in the BPA exposure group (median 1.075% vs. 0.54%, respectively; *p* = 0.041) compared to the control group. After potential confounders were adjusted for, subjects who were in the lowest, middle, and top tertiles of detected BPA had higher 5hmC rates than those with undetected BPA (1.11%, 1.15%, 0.945% vs. 0.52%). The differences were statistically significant except for the top tertile group (Table 3).

The linear regression analysis using the continuous BPA level found that increasing BPA was associated with increased 5hmC, but the association was marginally statistically significant (β = 0.046, 95%CI: 0.000, 0.0092). When the analyses were repeated within each tertile of urine BPA levels and in the control group, we observed statistically significant linear associations between urine BPA levels and 5hmC among those in the middle tertile group (β = 0.453, 95%CI: 0.070, 0.835) and those in the control group (no occupational BPA exposure) (β = 0.144, 95%CI: 0.003, 0.285) despite the markedly reduced sample size (Table 4).

## 4. Discussion

In the present study, we provide the first epidemiological evidence that BPA exposure is associated with increased 5hmC of the sperm *ACHE* gene in men. The linear association is also observed among subjects without occupational BPA exposure.

Our findings are comparable to those found in previous studies that focused on 5hmC changes in response to endo- and exogenous factors [42,43]. Sanchezguerra et al. recently found that exposure to ambient PM10 could affect the blood genomic content of 5hmC [42].

An increasing number of studies have suggested that BPA may affect human health by epigenetic regulation [13,19,44,45]. The emerging epigenetic modification, 5hmC, has been shown to play a role in regulating gene function during the differentiation of spermatogenic cells [46,47]. There is evidence that *ACHE* overexpression or activity is detected in apoptotic cells after the induction of apoptosis by different stimuli in many different type cells, such as hematopoietic stem cells, human neuroblastoma cells, and other cell lines [29,30,48]. The molecular mechanism underlying these effects is likely that ACHE promotes caspase-9 activation to increase nuclear condensation and polymerase cleavage [29,49,50,51]. Some studies reported that the transcriptional activity of the gene could be enhanced by DNA hypomethylation [46], which indicates that DNA hydroxymethylation in the *ACHE* promoter raises the gene activity. Abnormal *ACHE* expression is reported to be associated with sperm counts and motility [31]. This is supported by a recent report that the 5hmC rate of the sperm *ACHE* gene is higher in asthenozoospermia and oligoasthenozoospermia men than in normozoospermia men [52]. In our study, the higher rate of 5hmC of the sperm *ACHE* gene in subjects was associated with higher BPA exposure. This suggests the presence of upregulation of ACHE activity in the sperm from BPA-exposed men, which may eventually contribute to poor sperm concentration in the BPA-exposed men whom we had observed [12]. Our result is also in line with our previous report on the genome-wide upregulation of DNA 5′-hydroxymethylation in the spermatozoa of men exposed to BPA [23]. However, further studies need to be done to understand the changes in sperm quality across 5hmC levels.

A growing body of literature has reported adverse effects following developmental exposure to low doses of BPA [53,54]. Male rodents exposed to low levels of BPA displayed health impacts including altered serum testosterone levels and sperm quality [54,55]. In the present study, we did not observe a stronger association in the subgroup with the highest BPA exposure. In addition, we observed a similar association among the unexposed group. This non-monotonic effect is consistent with previous reports [53,56,57], in which the biological effect of BPA is not always stronger in high dose exposure than low dose exposure.

The relationship between environmental exposures and 5hmC of the sperm *ACHE* gene has not been reported previously. Our findings may have implications for understanding environmental exposure-induced male infertility via measuring 5hmC of the sperm *ACHE* gene. Measuring the levels of 5hmC in semen specimens rather than other tissues to assess the potential epigenetic effect of BPA on semen quality is straight-forward and accurate. However, there are some limitations in our study. First, we did not investigate sperm *ACHE* gene expression in this study because of the limited amount of individual semen samples. Next, we only detected one CpG site near the transcription start site (TSS) in the *ACHE* promoter region, which could not fully describe the relationship between the 5hmC of *ACHE* and its transcript expression level. Third, the high occupational exposure has restricted the generalizability of the present study. BPA concentrations of some subjects in the top tertile group in the present study are much higher than those reported in the USA [58]. However, in the present study, we did not observe a stronger association in the subgroup with the highest BPA exposure. It is less likely that the observed association can be explained by the extremely high exposure. Fourth, the levels of BPA were measured based on a single spot urine sample (the unexposed group) or samples pre and post one shift (the exposed group), which may not be perfect surrogates of long-term BPA exposure, thus attenuating the observed association due to non-differential misclassification. However, it has been reported that a single urine sample can be relatively representative of exposure within a certain period [59]. Finally, similar to many studies collecting biological specimens, participants’ refusal to provide biological samples (urine and/or semen) may lead to potential participation bias. However, since subjects in our study were not aware of both their BPA exposure and sperm *ACHE* hydroxymethylation levels before the biosamples were collected, the refusals may be considered random or undifferentiated in relation to exposure and outcome measures. Therefore, it seems unlikely that the observed association could be explained by participation bias.

## 5. Conclusions

In conclusion, our results indicate that 5hmC of the sperm *ACHE* gene is positively associated with BPA exposure. This provides supportive evidence for the effects of BPA on male fertility and other environmental exposure-related diseases.

## Figures and Tables

**Table 1 ijerph-16-00152-t001:** Distribution of urine bisphenol A (BPA) by characteristics of the study population (μg/g Cr).

Characteristics		BPA (μg/g Cr)						
	N (157)	GM (GSD)	Median (Q1, Q3)	Minimum	Maximum	5th	95th	*p* Value ^cd^
Age (years)								
<29	55	10.91 (32.94)	7.90 (LOD, 61.00)	LOD	109,600.22	LOD	22,991.39	0.81
30–35	39	10.60 (17.55)	11.67 (1.45, 43.01)	LOD	22,620.79	LOD	20,983.27	
>36	63	7.43 (54.55)	1.12 (LOD, 614.93)	0.16	264,219.38	LOD	13,418.01	
Education								
≤Middle school	41	35.94 (116.96)	28.52 (LOD, 3257.32)	LOD	264,219.38	LOD	101,203.06	<0.01
High school	86	7.59 (20.75)	9.41 (LOD, 55.54)	0.16	22,620.79	LOD	1918.31	
≥College	30	2.60 (9.89)	2.91 (LOD, 21.18)	LOD	1411.29	LOD	1010.85	
History of disease ^a^								
No	127	9.57 (36.07)	8.44 (LOD, 125.22)	0.16	264,219.38	LOD	21,965.79	0.83
Yes	30	8.18 (31.94)	8.80 (LOD, 115.08)	LOD	3723.77	LOD	8385.03	
Smoking								
No	51	5.31 (38.62)	1.24 (LOD, 282.81)	0.16	109,600.22	LOD	12,209.85	0.17
Yes	106	12.15 (32.71)	12.48 (LOD, 74.69)	LOD	264,219.38	LOD	19,048.45	
Alcohol consumption								
No	119	10.77 (37.53)	9.99 (LOD, 232.80)	0.16	264,219.38	LOD	22,620.79	0.36
Yes	38	5.84 (27.52)	6.48 (LOD, 67.96)	LOD	3723.77	LOD	3373.04	
Occupational exposure to BPA								
No	83	0.77 (6.33)	0.22 (LOD, 5.63)	0.16	74.5	LOD	23.11	<0.01
Yes	74	199.13 (19.65)	180.59 (23.39,1928.01)	0.74	264,219.38	1.45	23,979.51	
Categories by urine BPA level								
BPA undetected (below LOD) ^b^	53	LOD	LOD (LOD, LOD)	LOD	LOD	LOD	LOD	<0.01
Low tertile (LOD-13.84)	35	3.93 (2.96)	5.63 (1.55, 8.82)	0.16	13.84	0.36	13.44	
Middle tertile (13.84–274.83)	35	42.58 (2.26)	32.90 (22.78, 74.50)	15.15	274.84	15.39	250.78	
Top tertile (>274.83)	34	2937.79 (5.48)	2161.23 (683.21, 9771.63)	282.81	264,219.38	353.49	148,255.01	

^a^: Disease refers to any acute or chronic disease of the liver, kidney, or other organs. ^b^: The urine BPA values of undetectable were input as LOD/√2. ^c^: The urine BPA concentrations were natural log (ln) transformed. ^d^: Independent *t*-test or one-way ANOVA analysis. GM: Geometrical Mean. GSD: Geometric Standard Deviation. LOD: Limit of detection.

**Table 2 ijerph-16-00152-t002:** Distribution of the 5-hydroxymethylcytosine (5hmC) rate of the sperm acetylcholinesterase (*ACHE*) gene by characteristics of the study population and BPA exposure.

Groups		5hmC Rate of Sperm *ACHE* Gene					
	N	5th%	25th%		50th%	75th%	95th%
Occupational exposure to BPA							
No	83	0.12	0.27		0.54	1.35	2.41
Yes	74	0.11	0.43		1.075	1.78	2.57
Categories by urine BPA level							
BPA undetected	53	0.11	0.22		0.52	1	1.81
Low tertile (LOD-13.84)	35	0.15	0.44		1.11	2.07	2.49
Middle tertile (13.84–274.83)	35	0.13	0.48		1.15	2.03	2.61
Top tertile (>274.83)	34	0.1	0.23		0.945	1.61	2.57
Age (years)							
<29	55	0.11	0.36		0.58	1.74	2.43
30–35	39	0.11	0.36		0.84	1.69	2.61
>36	63	0.12	0.31		0.72	1.49	2.46
Smoking							
No	51	0.13	0.33		0.58	1.19	2.49
Yes	106	0.11	0.37		0.75	1.66	2.56
Alcohol consumption							
No	119	0.11	0.36		0.84	1.61	2.56
Yes	38	0.11	0.29		0.55	1.48	3.01
History of disease							
No	127	0.13	0.31		0.63	1.51	2.43
Yes	30	0.05	0.43		0.93	1.81	2.61

**Table 3 ijerph-16-00152-t003:** Linear regression of BPA exposure and the 5hmC rate of the *ACHE* gene.

Groups		5hmC Rate of Sperm *ACHE* Gene ^a^			
	N	Crude β(95%CI) ^b^	Crude *p*-Value	Adjusted β ^bc^ (95%CI)	*p*-Value
Occupational exposure to BPA					
No	83	Ref		Ref	
Yes	74	0.337 (0.019, 0.655)	0.038	0.336 (0.014, 0.657)	0.041
Categories by urine BPA level					
BPA undetected	53	Ref		Ref	-
Low tertile (LOD-13.84)	35	0.642 (0.218, 1.065)	0.003	0.661 (0.220, 1.102)	0.004
Middle tertile (13.84–274.83)	35	0.67 (0.246, 1.093)	0.002	0.682 (0.237, 1.127)	0.003
Top tertile (>274.83)	34	0.276 (−0.151, 0.703)	0.204	0.274 (−0.160, 0.708)	0.213
Age (years) ^d^					
<29	55	Ref		Ref	
30–35	39	0.106 (−0.317, 0.528)	0.622	0.032 (−0.395, 0.458)	0.883
>36	63	−0.019 (−0.392, 0.353)	0.919	−0.036 (−0.411, 0.339)	0.850
Smoking ^d^					
No	51	Ref		Ref	
Yes	106	0.141 (−0.2, 0.48)	0.416	0.176 (−0.178, 0.531)	0.327
Alcohol consumption ^d^					
No	119	Ref		Ref	
Yes	38	−0.124 (−0.499, 0.251)	0.515	−0.141 (−0.529, 0.247)	0.473
History of disease ^d^					
No	127	Ref		Ref	
Yes	30	0.064 (−0.34, 0.47)	0.755	0.063 (−0.348, 0.475)	0.762

^a^: Rates of 5hmC were natural log (ln) transformed. ^b^: Adjusted for age, history of disease, smoking, and alcohol consumption.^c^: Urine BPA were categorically variable. ^d^: Analyzed in occupational exposure to BPA.

**Table 4 ijerph-16-00152-t004:** Linear regression of BPA exposure and the 5hmC rate of the sperm *ACHE* gene ^a^.

Groups	N	Crude β (95%CI)	Crude *p*-Value	Adjusted β ^b^ (95%CI)	*p*-Value
All subjects	157	0.048 (0.003, 0.093)	0.036	0.046 (0.000, 0.092)	0.051
Low tertile (LOD-13.84)	35	0.068 (−0.229, 0.366)	0.642	0.104 (−0.231, 0.440)	0.529
Middle tertile (13.84–274.83)	35	0.402 (0.025, 0.779)	0.002	0.453 (0.070, 0.835)	0.021
Top tertile (>274.83)	34	0.160 (−0.051, 0.37)	0.133	0.137 (−0.097, 0.370)	0.242
Non-occupational exposure	83	0.127 (0.01, 0.244)	0.033	0.144 (0.003, 0.285)	0.045
Age (years) ^c^					
<29	55				
30–35	39	0.105 (−0.317, 0.528)	0.622	0.087 (−0.337, 0.512)	0.685
>36	63	−0.019 (−0.391, 0.353)	0.919	−0.005 (−0.38, 0.371)	0.981
Smoking ^c^					
No	51				
Yes	106	0.141 (−0.2, 0.48)	0.416	0.122 (−0.236, 0.48)	0.501
Alcohol consumption ^c^					
No	119				
Yes	38	−0.123 (−0.499, 0.251)	0.515	−0.122 (−0.511, 0.268)	0.537
History of disease ^c^					
No	127				
Yes	30	0.064 (−0.34, 0.47)	0.755	0.065 (−0.347, 0.477)	0.756

^a^: Rates of 5hmC and urine BPA concentrations were natural log (ln) transformed. ^b^: Adjusted for age, history of disease, smoking, and alcohol consumption. ^c^: Analyzed in all subjects.

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
