# Peer review of "Bisphenol A Exposure and Sperm ACHE Hydroxymethylation in Men"

_ijerph, 2019, doi:10.3390/ijerph16010152_

Round 1

Reviewer 1 Report

This is a valuable article but the following questions should be clarified before considering publication.

1.Two urine samples (pre-shift and post-shift) were collected from each male worker of the BPA-exposed group. Please compare the differences of pre-shift and post-shift BPA levels. In addition, please describe the detection rate of pre-shift and post-shift BPA levels.

 2.In the materials and methods, 5-mC and 5-hmC were analyzed and quantitated using EpiMark 5-hmC and 5-mC Analysis Kit.  But there is no results about 5-mc in the text.

3.Please describe minimum, maximum, and detection rate of urinary BPA levels in exposure group and control group, respectively. And we can understand complete distribution of urinary BPA levels.

4.No descriptions of the statistical methods are presented on Table1. Please write the statistical methods in the text and Table 1.

5.Please add minimum, maximum, 5th, and 95th percentiles in Table 1.

6.In Table 1, post hoc test must be further used in the result of education.

7.Please correct the “GM(std)” to GM(GSD). The geometric standard deviation (GSD) describes how spread out are a set of numbers whose preferred average is the geometric mean.

8.Please cite the references in page3 lines 119-121.  It is important to cite these references.

9.Please show unadjusted b, 95% CI and p value of BPA exposure in Table2 and Table 3. This can help us understand the changes before and after adjustment.

10.Please also show the unadjusted and adjusted b, 95%CI and p-value of age, smoking, alcohol consumption, history of disease in Table2 and Table3.

11.It is very important to clarify whether there is collinearity in the regression analysis.

12.Please indicate the urinary BPA levels and rates of 5hmc in Table 2 and Table 3 are log transformed data, respectively.

13.Page 2 line 89. Please briefly describe the conditions and procedures of the analysis.

Author Response

1.Two urine samples (pre-shift and post-shift) were collected from each male worker of the BPA-exposed group. Please compare the differences of pre-shift and post-shift BPA levels. In addition, please describe the detection rate of pre-shift and post-shift BPA levels.

Response: We have evaluated the differences of BPA levels between pre-shift and post-shift, the result of which is presented in supplemental table (Table S1). The creatinine-corrected BPA concentrations were natural log (ln) transformed to achieve normal distribution to Paired-Samples T Test analysis.  BPA levels were higher in post-shift urine samples.

Table S1 The differences of pre-shift and post-shift BPA levels in BPA-exposed group

percentiles

N

detection rate%

Median

minimum

maximum

5th

95th

t

p

Pre-shift

74

96.3

89.96

LOD

217060.90

0.57

12862.53

Post-shift

74

100

180.59

0.75

264219.40

1.45

23979.51

3.62

0.001

2. In the materials and methods, 5-mC and 5-hmC were analyzed and quantitated using EpiMark 5-hmC and 5-mC Analysis Kit.  But there is no results about 5-mc in the text.

Response: We apologized for the unclear description. EpiMark 5-hmC and 5-mC Analysis Kit can be used to analyze and quantitate 5-methylcytosine and 5-hydroxymethylcytosine within a specific locus. Only 5-hmC was recorded in the present study. We have revised the materials and methods as “5-hmC was analyzed and quantitated using EpiMark 5-hmC and 5-mC Analysis Kit (NEB, #E3317S) according to the manufacturer’s protocols.”(page3, line 104-105)

3.Please describe minimum, maximum, and detection rate of urinary BPA levels in exposure group and control group, respectively. And we can understand complete distribution of urinary BPA levels.

Response: We thank you for your suggestion and we have added minimum, maximum, 5th, and 95th percentiles in table1(page4,5), added detection rate of urinary BPA levels in exposure group and control group in result as” Detection rate of urinary BPA levels in exposure group and control group was 100% and 36.14%, respectively.”(page3 , line 138)

4.No descriptions of the statistical methods are presented on Table1. Please write the statistical methods in the text and Table 1.

Response: We have added the statistical methods in the text as ” The urine BPA concentrations were natural log (ln) transformed to achieve normal distribution and then independent t-test or one-way ANOVA analysis was performed to analyze differences of BPA levels” (page3 line 118-119 )and added the table annotation in table1 as ”c: The urine BPA concentrations were natural log (ln) transformed; d: independent t-test or one-way ANOVA analysis.”(page5)

5.Please add minimum, maximum, 5th, and 95th percentiles in Table 1.

Response: We have added minimum, maximum, 5th, and 95th percentiles in table1(page4).

6.In Table 1, post hoc test must be further used in the result of education.

Response: We have added the result of post hoc test in education in the text as” Compared to men with middle school or below, BPA levels in those with high school and with college and above were relatively lower (Table S2).” (page4 line 142-143). The results of post hoc test in education were shown as supplemental table (Table S2):

Table S2 The difference of urine BPA in men with different education category

(I) edu

(J) edu

Mean Difference (I-J)

standard error

p

  Middle school

High school

1.56*

0.66

0.019

College

2.63*

0.83

0.002

High school

Middle school

-1.56*

0.66

0.019

College

1.07

0.73

0.146

College

Middle school

-2.63*

0.83

0.002

High school

-1.07

0.73

0.146

*α=0.05

7.Please correct the “GM(std)” to GM(GSD). The geometric standard deviation (GSD) describes how spread out are a set of numbers whose preferred average is the geometric mean.

Response: We apologized for the mistakes. We have corrected the “GM(std)” to GM(GSD).

8.Please cite the references in page3 lines 119-121.  It is important to cite these references.

Response: We have added the references in the text.(page3)

9.Please show unadjusted b, 95% CI and p value of BPA exposure in Table2 and Table 3. This can help us understand the changes before and after adjustment.

Response: We have added crudeβ(95%CI) of BPA exposure (Table3 and Table4, in revised manuscript ) (page6 and 7,respectively). After adding crudeβ(95%CI) to the original table2, the table was a little too big, and we thus divided it into two tables Table2(page5) and Table3(page6).

10.Please also show the unadjusted and adjusted b, 95%CI and p-value of age, smoking, alcohol consumption, history of disease in Table2 and Table3.

Response: We have added crude and adjustedβ(95%CI) and p-value of age, smoking, alcohol consumption, history of disease in Table3 (page6)  and Table4(page7).

11. It is very important to clarify whether there is collinearity in the regression analysis.

Response: Before multivariate linear regression, we made a diagnosis of collinearity between variables, the results was shown in supplemental table (Table s3). The VIF of each variable is less than 10, there is no multicollinearity in the model.

Table S3 Collinearity diagnostics

variables

BPA

age

smoke

alcohol

disease

1/VIF

0.976

0.938

0.925

0.933

0.945

VIF

1.025

1.066

1.081

1.071

1.058

12.Please indicate the urinary BPA levels and rates of 5hmc in Table 2 and Table 3 are log transformed data, respectively.

Response: We have added the table annotation as” c: Rates of 5hmc were natural log (ln) transformed” in Table 3(page6 ) and “a: Rates of 5hmc and urine BPA concentrations were natural log (ln) transformed” in Table4(page7).

13.Page 2 line 89. Please briefly describe the conditions and procedures of the analysis.

Response: We have added a brief description in the Method Section as “Briefly, urine samples were treated with phosphorousacid buffer/ β-glucuronidase for hydrolyzation, and were subsequently extracted twice usingether (HPLC grade, Dikma). The supernatants were collected and evaporated with nitrogengas. The residue was dissolved in 60% acetonitrile and analyzed using HPLC equipment.”(page2 line91-94)

Reviewer 2 Report

The manuscript deals with the assessment of BPA exposure with hydroxymethylation level of sperm ACHE gene that is potentially related to male fertility, thus representing a crucial issue in the reproductive effects by endocrine disrupting chemicals. The article is well written and deserves to be published. I point out only some minor revisions.

ABSTRACT

At lines 26-27, the statement is not completely correct, since a linear association was also reported in the middle tertile group.

INTRODUCTION

At line 46, consider to write full name of non-LTR retrotransposons

At line 51 The statement “Recent animal study reported that exposure to BPA 51 inhibits global DNA hydroxymethylation in the adult testis….” appears to be in contrast with what reported at lines 53-54. Could the authors better explain this?

DISCUSSION:

At line 200 the authors stated that BPA exposure was based on a single urine measure. However, in methods, they wrote that two urine samples were collected from the exposed men, and one from the control group. May they elucidate this point?

At lines 204-206, the sentence is rather confused, consider to rephrase it.

The statement at line 206-207 could be linked to that at line 200.

Author Response

ABSTRACT

1.At lines 26-27, the statement is not completely correct, since a linear association was also reported in the middle tertile group.

Response: We thank you for your suggestion. we have revised the manuscript as ” In linear regression analysis adjusted for confounders, a  positive linear association between urine BPA concentration and 5hmc rate of sperm ACHE gene was observed, although the association did not  reach statistical significance in all categories after stratified by BPA tertile.”(page1,line 25-28)

INTRODUCTION

2. At line 46, consider to write full name of non-LTR retrotransposons

Response: We have added the brief description in the text as “the most abundant and the only active autonomous non-LTR retrotransposons (retrotransposon that do not contain long terminal repeats ) in the human genome”.(page2,line47-48)

3. At line 51 The statement “Recent animal study reported that exposure to BPA 51 inhibits global DNA hydroxymethylation in the adult testis….” appears to be in contrast with what reported at lines 53-54. Could the authors better explain this?

Response: We apologized for the unclear description. As described in the literature, DNA hydroxymethylation is species specific and tissue-specific, and it is different in the different loci of genes. In our manuscript, we tried to elaborate the fact that the 5-hmC decreased with BPA exposure in gonads of adult zebrafish Danio rerio while increased in the sperm of human, which is reasonable. We want to support the hypothesis the 5-hmC is considered one of the key molecules involved in the process of DNA demethylation caused by BPA.

DISCUSSION:

4. At line 200 the authors stated that BPA exposure was based on a single urine measure. However, in methods, they wrote that two urine samples were collected from the exposed men, and one from the control group. May they elucidate this point?

Response: We have revised the statement in the text as ” Fourth, the levels of BPA in the men were measured based on a single spot urine sample (the unexposed group )or samples pre and post one shift (the exposed group ), which may not be perfect surrogates of long-term BPA exposure, thus attenuating the observed association due to non-differential mis-classification. However, it has been reported that a single urine sample can be relatively representative of exposure within a certain period.”(page9, line 226-231)

5. At lines 204-206, the sentence is rather confused, consider to rephrase it.

Response: We apologized for this. We have revised the statement in the text as” Finally, like many studies collecting biological specimens, participants’ refusal to provide biological samples (urine and/or semen) may lead to potential participation bias. However, since subjects in our study were not aware of both their the urine BPA levels and sperm ACHE hydroxymethylation levels before the biosamples were collected, the refusals may be considered random or undifferentiated in relation to exposure and outcome measures. Therefore, it seemed unlikely that the observed association could be explained by participation bias.” (page9, line 232-238)

6. The statement at line 206-207 could be linked to that at line 200.

Response: We have deleted the statement of line 206-207.

Reviewer 3 Report

The authors Xiuxia Song et al., have submitted a paper entitled “ Bisphenol A Exposure and Sperm ACHE Hydroxymethylation in Men” in Int. J. Environ. Res. Public Health ,for publication.

The exposure to BPA has a disrupting potential endocrine effects with higher risks of health in organism, particularly,in male reproductive function in human.In the present study an increased 5hmc of sperm ACHE gene in men is reported related to the epigenetic DNA modification. In fact, 5hmc, has been shown toplay a role in regulating gene function during the differentiation of spermatogenic cells. In conclusion, the results indicated that 5hmc of sperm ACHE gene was positively associated with BPA exposure.

I suggest the publication of this manuscript.

Author Response

We appreciate your recognition of the present study.

Round 2

Reviewer 1 Report

The BPA concentration can be as high as 264219 μg/g cr. ?  The extreme outliers in the BPA levels of the exposed group may cause significant correlations or differences in the results of this study.  Please refer the following article. (Hines, C. J., Jackson, M. V., …..(2017). Urinary bisphenol A (BPA) concentrations among workers in industries that manufacture and use BPA in the USA. Annals of work exposures and health, 61(2), 164-182.)

The unit of LOD is wrong (page 3, line 95).  Is there any wrong calculation of BPA concentration due to the wrong unit? Please check it.

The SD is not equal to GSD. The original data of SD has not been modified. If the GSD is not calculated, please delete it and just show GM.

The study population and method of the present article is similar to the content of another article (Association of Bisphenol A Exposure with LINE-1 Hydroxymethylation in Human Semen. Int. J. Environ. Res. Public Health 2018, 15(8), 1770). Please explain the differences betwwen the two articles. 

Author Response

1. The BPA concentration can be as high as 264219 μg/g cr. ?  The extreme outliers in the BPA levels of the exposed group may cause significant correlations or differences in the results of this study.  Please refer the following article. (Hines, C. J., Jackson, M. V., …..(2017). Urinary bisphenol A (BPA) concentrations among workers in industries that manufacture and use BPA in the USA. Annals of work exposures and health61(2), 164-182.)

Response: We thank you for your suggestion. BPA concentrations of some subjects in the top tertile group in our study are much higher compared to those reported in the article(Hines, C. J., et al. , 2017). However, the 5hmc rate of sperm ACHE gene was higher in the top tertile of detected BPA group compared to that in undetected group(1.03%vs 0.68%),while it is lower compared to that in lowest, middle tertile BPA group (1.03%vs 1.26%,1.33%). We did not observe a stronger association in the subgroup with highest BPA exposure (Table4 page6). So, the extreme outliers in the BPA levels of the exposed group may weaken the correlation between BPA and 5hmc. Our results may underestimate the relationship of BPA and 5hmc.

To rule out the effect of the extreme outliers in the BPA levels of the exposed group, we repeated the linear regression analysis after excluding the top tertile BPA group. We found the association between urine BPA levels and 5hmc was strengthened: adjustedβ95%CI changed from 0.046(0.000, 0.092)( Table4, page6) to 0.160(0.081, 0.242). 

We have added the statement in the limits as” Third, the high occupational exposure restricted the generalizability of present study. BPA concentrations of some subjects in the top tertile group in present study were much higher compared to those reported in the USA(Hines CJ, et al. , 2017). However, in the present study, we did not observe a stronger association in the subgroup with highest BPA exposure. It is less likely that the observed association can be explained by the extremely high exposure.”(page8,line220-223)

2. The unit of LOD is wrong (page 3, line 95).  Is there any wrong calculation of BPA concentration due to the wrong unit? Please check it.

Response: We apologized for the mistake. We have revised the unit of LOD in the text (page 3,line95). It was a typo error when we were preparing our original manuscript, the data in manuscript is base on the right unit.

3. The SD is not equal to GSD. The original data of SD has not been modified. If the GSD is not calculated, please delete it and just show GM.

Response: We apologized for the mistake in our original manuscript using the std. It was a typo error, the original data in manuscript actually is GSD.

4. The study population and method of the present article is similar to the content of another article (Association of Bisphenol A Exposure with LINE-1 Hydroxymethylation in Human Semen. Int. J. Environ. Res. Public Health 201815(8), 1770). Please explain the differences between the two articles. 

Response: The study population of the two articles was selected from a same prospective cohort study examining the health effect of occupational BPA exposure in several regions of China. Both of the two articles were to explore the relationship of BPA exposure and 5hmc of sperm DNA and intended to provide supportive evidence for BPA’s effects on male fertility. Because long interspersed element-1 (LINE-1), comprised of approximately 17% of human genomic DNA, is the most abundant and the only active autonomous non-LTR retrotransposon in human genome. We first selected LINE-1 to verify the hypothesis. However, some studies showed ACHE gene is sperm-expressed and participate in spermatogenesis. We hypothesized that ACHE gene may be the target gene of BPA in sperm. Both of the two articles have supported the hypothesis that the alteration of 5-hmC in sperm gene caused by BPA may be the mechanism underlying BPA-induced adverse effect on male reproductive function.